# In Silico RNAseq and Biochemical Analyses of Glucose-6-Phosphate Dehydrogenase (G6PDH) from Sweet Pepper Fruits: Involvement of Nitric Oxide (NO) in Ripening and Modulation

**DOI:** 10.3390/plants12193408

**Published:** 2023-09-27

**Authors:** María A. Muñoz-Vargas, Salvador González-Gordo, Jorge Taboada, José M. Palma, Francisco J. Corpas

**Affiliations:** Group of Antioxidants, Free Radicals and Nitric Oxide in Biotechnology, Food and Agriculture, Department of Stress, Development and Signaling in Plants, Estación Experimental del Zaidín (Spanish National Research Council, CSIC), C/Profesor Albareda, 1, 18008 Granada, Spain; mangeles.munoz@eez.csic.es (M.A.M.-V.); salvador.gonzalez@eez.csic.es (S.G.-G.); jtaboada@correo.ugr.es (J.T.)

**Keywords:** enzyme activity, fruit ripening, hydrogen sulfide, molecular modeling, NADPH, NADP dehydrogenases, pentose phosphate pathway

## Abstract

Pepper (*Capsicum annuum* L.) fruit is a horticultural product consumed worldwide which has great nutritional and economic relevance. Besides the phenotypical changes that pepper fruit undergo during ripening, there are many associated modifications at transcriptomic, proteomic, biochemical, and metabolic levels. Nitric oxide (NO) is a recognized signal molecule that can exert regulatory functions in diverse plant processes including fruit ripening, but the relevance of NADPH as a fingerprinting of the crop physiology including ripening has also been proposed. Glucose-6-phosphate dehydrogenase (G6PDH) is the first and rate-limiting enzyme of the oxidative phase of the pentose phosphate pathway (oxiPPP) with the capacity to generate NADPH. Thus far, the available information on G6PDH and other NADPH-generating enzymatic systems in pepper plants, and their expression during the ripening of sweet pepper fruit, is very scarce. Therefore, an analysis at the transcriptomic, molecular and functional levels of the G6PDH system has been accomplished in this work for the first time. Based on a data-mining approach to the pepper genome and fruit transcriptome (RNA-seq), four *G6PDH* genes were identified in pepper plants and designated *CaG6PDH1* to *CaG6PDH4*, with all of them also being expressed in fruits. While *CaG6PDH1* encodes a cytosolic isozyme, the other genes code for plastid isozymes. The time-course expression analysis of these *CaG6PDH* genes during different fruit ripening stages, including green immature (G), breaking point (BP), and red ripe (R), showed that they were differentially modulated. Thus, while *CaG6PDH*2 and *CaG6PDH4* were upregulated at ripening, *CaG6PDH*1 was downregulated, and *CaG6PDH3* was slightly affected. Exogenous treatment of fruits with NO gas triggered the downregulation of *CaG6PDH2*, whereas the other genes were positively regulated. In-gel analysis using non-denaturing PAGE of a 50–75% ammonium-sulfate-enriched protein fraction from pepper fruits allowed for identifying two isozymes designated CaG6PDH I and CaG6PDH II, according to their electrophoretic mobility. In order to test the potential modulation of such pepper G6PDH isozymes, in vitro analyses of green pepper fruit samples in the presence of different compounds including NO donors (*S*-nitrosoglutathione and nitrosocysteine), peroxynitrite (ONOO^−^), a hydrogen sulfide (H_2_S) donor (NaHS, sodium hydrosulfide), and reducing agents such as reduced glutathione (GSH) and L-cysteine (L-Cys) were assayed. While peroxynitrite and the reducing compounds provoked a partial inhibition of one or both isoenzymes, NaHS exerted 100% inhibition of the two CaG6PDHs. Taken together these data provide the first data on the modulation of CaG6PDHs at gene and activity levels which occur in pepper fruit during ripening and after NO post-harvest treatment. As a consequence, this phenomenon may influence the NADPH availability for the redox homeostasis of the fruit and balance its active nitro-oxidative metabolism throughout the ripening process.

## 1. Introduction

Glucose-6-phosphate dehydrogenase (G6PDH; EC 1.1.1.49) catalyzes the conversion of glucose-6-phosphate (G6P) to 6-phosphogluconolactone with the concomitant generation of NADPH (reduced nicotinamide adenine dinucleotide phosphate), which is considered the rate-limiting enzyme of the oxidative phase of the pentose phosphate pathway (oxiPPP). There are two additional enzymes in this route, the 6-phosphogluconolactonase (6PGL, EC 3.1.1.31) and 6-phosphogluconate dehydrogenase (6PGDH, EC 1.1.1.44), with this latter also generating NADPH, thus contributing, together with the photosynthetic ferredoxin-NADP^+^ reductase (FNR), to the NADPH pool in plants. This cofactor is required in numerous biosynthetic pathways as well as to support the metabolism of reactive oxygen species (ROS). Recently, NADPH has been proposed as a quality footprinting for the marketability of horticultural crops [1].

Regarding G6PDH, it has a wide distribution in living organisms from bacteria to mammals and its function has been related to numerous developmental processes as well as in the response to environmental stresses [2]. In humans, G6DPH deficiency is related to multiple diseases and pathologies affecting the cardiovascular system [3], but it is also involved in malaria [4] or cancer episodes, among others [5].

In higher plants, G6PDHs are made up of a family of genes that code for isoenzymes located in plastids [6], cytosol [7,8], peroxisomes [9], and in the endoplasmic reticulum [10]. Additionally, it has been reported that some G6PDH isozymes seem to have their own redox regulation. Thus, it has been shown that plastid G6PDH is sensitive to reducing agents such as dithiothreitol (DTT), while the cytosolic and peroxisomal isozymes do not seem to be affected [6,9,11,12,13,14]. More recently, it has been shown that G6PDH activity can be also regulated by nitric oxide (NO)-mediated post-translational modifications (PTMs) such as *S*-nitrosation [15].

Different studies have correlated the G6PDH activity in plants with physiological processes such as germination and development [16,17,18,19,20], but also with the mechanisms of response to numerous environmental stresses [21,22] including salinity [23,24,25], water stress [22,23], mechanical wounding, arsenic, chromium, aluminum [24], herbicides [25], low temperature [26,27], potassium deficiency or pathogens [28,29].

The pepper (*Capsicum annuum* L.) is a fruit originally from Central and South America that was introduced in Europe by C. Columbus in 1493. At present, pepper is a relevant crop whose fruit is consumed worldwide either fresh, canned, or processed products, i.e., as a food condiment. It is characterized by containing a significant quantity of vitamin C, provitamin A, and minerals such as calcium, as well as other compounds with healthy properties [30]. During the ripening of pepper fruits, there is a functional reactive oxygen and nitrogen metabolism (ROS and RNS, respectively) which has been associated with a physiological nitro-oxidative stress, where different ROS-generating and antioxidant systems are differentially modulated. They include NADPH oxidase, superoxide dismutase, catalase, ascorbate peroxidase, and peroxidases, among others [31,32]. However, thus far, the available information on G6PDH and other NADPH-generating enzymatic systems in pepper plants and their expression during the ripening of sweet pepper fruit is very scarce.

Earlier studies found that G6PDH activity was modulated during pepper fruit ripening [33,34] but to our knowledge, there is no data on the putative genes that code for G6PDH in pepper fruits and their modulation. Therefore, the main aim of this study was to identify the *G6PDH* genes in sweet pepper fruits and how they are modulated during ripening and by the effect of the post-harvest exogenous application of NO. Likewise, this study has evaluated how the fruit G6PDH isozymes are affected by NO and H_2_S donors, as well as by reducing agents. Accordingly, an analysis at transcriptomic, molecular, and functional levels of the G6PDH system has been accomplished in this work for the first time. This approach may contribute to providing this enzymatic system with an essential function in the redox metabolism of pepper fruits in the scenario of considering NADPH as a marketability parameter.

## 2. Results

### 2.1. Identification and Analysis of the G6PDH Genes from Pepper (Capsicum annuum L.)

Through the evaluation in the databases of the available G6PDH sequences from several plant species, the mining of the *Capsicum annuum* L. genome allowed us to identify four *G6PDH* genes, designated *CaG6PDH1* to *CaG6PDH4*, according to their chromosomal distribution. Thus, each gene was located in either chromosome (Chr.) 2, 4, 7, or 8, respectively. On the other hand, data mining in the transcriptome previously obtained from sweet pepper fruits [31] indicated that these four genes were also expressed in other fruits. Table 1 summarizes some properties of these genes and their corresponding encoded G6PDH proteins including the number of amino acids (aa), molecular mass (kDa), and their putative subcellular localization, among other features.

The genomic organization of the four *CaG6PDH* genes in the four chromosomes was analyzed (Figure 1). *CaG6PDH1* contained 15 exons in Chr. 2, *CaG6PDH2* and *CaG6PDH3* comprised 10 exons in Chrs. 4 and 7, respectively, whereas *CaG6PDH4* contained 11 exons in Chr. 8. It is remarkable that the length of the introns was very different among the four *CaG6PDHs*, with *CaG6PDH3* being the gene with two very long introns expanding above 6000 bp each.

As part of the in silico analysis, the identification of cis-regulatory elements in 1500 upstream regions of the *CaG6PDH* genes was accomplished. Figure 2 depicts the Heatmap analysis of the 26 identified cis-regulatory elements grouped into four families. They include i. DNA, regulation, and cell cycle; ii. light responsive; iii. stress; and, iv. phytohormones. The cis-regulatory elements that exert the most remarkable effects (reddish squares) were Box4 and G-Box, as part of the light-responsive family, as well as ARE (essential for anaerobic induction) and LTR (long terminal repeat) related to stress.

### 2.2. G6PDH Proteins from Pepper: Sequence and Phylogenetic Analysis

The analysis of the protein sequences encoded by the identified *CaG6PDH* genes (Table 1) indicates that their molecular mass ranged from 58 kDa for the cytosolic isozyme and around 65–67 kDa for the plastid isozymes. Likewise, the primary structure of these CaG6PDHs and their alignment allowed us to identify the different conserved amino acid motifs. Figure 3 shows the protein sequence alignment of the four CaG6PDHs, where the amino acids involved in the binding sites of G6P (RIDHYLGK) and NADP^+^ (NEFVIRLQP) are indicated. In the same figure, the Rossmann fold (GASGDLA) is also highlighted, which corresponds to the tertiary fold present in proteins that bind nucleotides, such as FAD^+^, NAD^+^, and NADP^+^. Only the plastid CaG6PDH4 showed certain modifications in these motifs. Thus, Figure 4 illustrates the predicted tertiary structure of the cytosolic and plastid CaG6PDHs from pepper fruits, where the binding site of the substrate (G6P), the cofactor (NADP), the Rossmann fold, and the cysteine residues of the plastid isozymes involved in redox regulation are labeled.

Further analysis of the predicted CaG6PDH proteins was conducted. Thus, Figure 5 illustrates the phylogenetic comparison among the 53 G6PDHs reported from 15 plant species. Also, Appendix A summarizes the protein IDs of all the G6PDH isozymes used for this phylogenetic analysis. This assessment allowed the identification of four G6PDH groups, designated as I to IV, which are represented with different colors in Figure 5. Group I contains the cytosolic CaG6PDHs, while the other groups harbor the plastid CaG6PDHs. Interestingly, none of these G6PDHs seems to be associated with peroxisomes, although their presence in these organelles has been reported.

To gain a deeper knowledge of the dynamic metabolic processes among the four CaG6PDHs, the possible protein–protein interaction (PPI) network of these proteins was explored. Figure 6 displays the analysis of the predicted PPI network using the STRING database, version 11.0 (https://string-db.org/; accessed on 10 February 2023) which facilitates the visualization/evaluation of the functional association of these CaG6PDHs. This model suggests that plastid CaG6PDH4 has a central position since it is the one that displays the highest interaction score.

### 2.3. Fruit CaG6PDH Genes: Expression during Ripening and after Exogenous NO Treatment

The analysis, using the RNAseq of sweet pepper fruits, of the four *CaG6PDH* genes was conducted at the different ripening stages and after the exposure of the fruits to exogenous NO gas. Appendix A illustrates the experimental design, which included green immature (G), breaking point (BP1), and red ripe (R) pepper fruits. Furthermore, two additional groups were established: fruits treated with 5 ppm NO for 1 h (BP2 + NO) and another untreated group (BP2 − NO) which corresponded to the control group against the NO-treated fruits. Figure 7 shows the time-course analysis of the four genes identified in pepper fruits. *CaG6PDH2* and *CaG6PDH4* were upregulated during ripening whereas *CaG6PDH1* was downregulated and *CaG6PDH4* was slightly modified. Exogenous NO treatment of the fruit caused slight downregulation of *CaG6PDH2*, whereas *CaG6PDH1*, *CaG6PDH3*, and *CaG6PDH4* were upregulated to different degrees.

### 2.4. Identification of the CaG6PDH Isozymes in Pepper Fruits

As part of the biochemical characterization of the CaG6PDHs in pepper fruits, in-gel analyses of the isoenzymatic activity were accomplished by applying non-denaturing PAGE. Since preliminary assays showed very low activity in pepper fruits, an enriched 50–75% ammonium sulfate protein fraction from green pepper fruits was prepared and the total activity is shown in Figure 8a. Likewise, Figure 8b depicts the isoenzymatic pattern of the CaG6PDH detected in green pepper fruits after non-denaturing PAGE in 6% acrylamide gels. After loading about 200 μg protein of this enriched fraction onto wells, two isozymes were detected and were designated as CaG6PDH I and CaG6PDH II, according to their increasing electrophoretic mobility in the gels. Both isozymes represented 27% and 73% of the total activity, respectively.

The regulation of CaG6PDH isozymes by different modulating compounds was investigated. Thus, pepper fruit samples were preincubated with a battery of reagents including the peroxynitrite (ONOO^−^) donor SIN-1, as a nitrating compound; the NO donors and nitrosating agents GSNO (S-nitrosoglutathione) and CysNO (nitrosocysteine); the reductant L-cysteine, reduced glutathione (GSH); the H_2_S donor sodium hydrosulfide (NaHS); and potassium cyanide, as an inhibitor of key enzymes involved in the mitochondrial and ROS metabolism such as cytochrome c oxidase or CuZn superoxide dismutase, respectively. The quantification of how the two CaG6PDHs were affected by all these compounds after non-denaturing PAGE and in-gel analysis is depicted in Figure 8c. Peroxynitrite caused an 86% and 80% inhibition of CaG6PDH I and II, respectively. GSH totally inhibited CaG6PDH I, but CaG6PDH II was not affected. The GSNO did not appear to affect either of the two CaG6PDHs. Nonetheless, L-Cys provoked a highly significant inhibition of both CaG6PDHs, whereas CysNO did not affect the activity of any of the isozymes, thus suggesting that NO exerts a protective role against the inhibitory effect of the L-Cys. In the case of the H_2_S donor NaHS, it triggered a 100% inhibition of both CaG6PDH isozymes. Finally, cyanide did not affect the activity.

## 3. Discussion

Pepper fruits are of great agronomic importance and their ripening is modulated by multiple factors. Among them, the cellular redox state is of great relevance. Thus, NADPH is an essential cofactor that contributes to the redox homeostasis and the proper functionality of plant cells. In fact, its levels have been proposed as a quality marker for horticultural products [1]. Therefore, the analysis of G6PDH as the first NADP-generating enzyme of oxiPPP will provide new basic information on how this enzymatic system is regulated during the ripening of pepper fruits. Thus, following the expression level of the *CaG6PDH* genes during this physiological process, and how the identified isozymes are modulated by different compounds with recognized signaling properties, mainly NO and H_2_S [15,35], will contribute to understanding the role of the G6PDH system in the physiology and nutritional quality of pepper fruits.

### 3.1. The Pepper Genome Contains Four CaG6PDH Genes Which Are All Expressed in Fruits

The number of *G6PDH* genes changes significantly among plant species. Thus, 19 *G6PDH* genes in strawberry (*Fragaria × ananassa*) [18], 9 in soybean (*Glycine max*) [35], 6 in *Arabidopsis thaliana* [8], 5 in rice (*Oryza sativa*), *Arabidopsis tauchii*, tomato (*Solanum lycopersicum*) [27], and maize (*Zea mays*) [27], and 4 *G6PDH* genes in rubber tree (*Hevea brasiliensis*) [36] and einkorn wheat (*Triticum urartu*) [37] have been reported. Therefore, the four *CaG6PDH* genes identified in this work indicate that pepper is among the species with the lowest number of *G6PDH* genes.

The number and length of introns/exons in the four *G6PDH* genes were also highly variable. Between 10–15 exons were counted in the *G6PDH* genes and, in the case of the *CaG6PDH3* gene (almost 24 kb length), it contained two very large introns above 6 kb each. Maize plants enclose five genes and, among them, the *ZmG6PDH4* gene length is almost 18 kb with 9 exons and two large introns [27]. Strawberries have 19 *G6PDH* genes with 10 to 15 exons, but the largest gene which corresponds to *FaG6PD14* has less than 7 kb [18]. All these examples with many introns/exons with different lengths indicate the great diversity of the *G6PDH* gene structures in higher plants. This eventuality makes them prone to being differently modulated and expressed according to the surrounding conditions. Usually, the genes with the highest number of introns are the most recent in evolution, and also have the greatest regulation. Thus, the intron sequences could exert several functions such as enhancing gene expression, regulating alternative splicing, and controlling mRNA transport and chromatin assembly, among others [38,39,40]. According to this, *CaG6PDH1* is possibly the gene with the greatest chance of being modulated, but this issue needs to be confirmed with further research.

The structural analysis of the four CaG6PDH-encoded proteins shows the conserved domain for the substrate (G6P) and the cofactor (NADP), as well as the Rossmann fold. This last domain is an additional tertiary fold with the capacity to bind NADP and is present in other proteins that bind different nucleotides such as FAD and NAD [41]. Furthermore, there are two cysteine residues conserved in the three plastidial CaG6PDH isozymes. In Arabidopsis, these Cys residues correspond to the Cys212 and Cys220 of the chloroplast AtG6PDH4 which are also essential for interacting with AtG6PDH1, allowing this latter protein to be imported into the peroxisome [42]. Thus, it has been shown in Arabidopsis that, among the six AtG6PDHs, none of them contains the canonical peroxisomal targeting signals (PTSs), of either type 1 or 2. However, the chloroplastic AtG6PDH1 has a putative PTS motif near the C-terminus, which interacts through the Cys212 and Cys220 residues with the chloroplastic AtG6PDH4 isozyme. As a consequence, this interaction allows an unusual mechanism of import of the G6PDH into peroxisomes as a G6PDH4–G6PDH1 heteromer [42]. Within the CaG6PDHs from pepper fruit, these cysteine residues have been identified in all the plastid isozymes (CaG6PDH2, CaG6PDH3, and CaG6PDH4), but not in the cytosolic CaG6PDH1. This suggests that this importing mechanism may also operate in pepper fruit peroxisomes since G6PDH activity was detected in these organelles isolated from pepper fruits [43].

### 3.2. During Fruit Ripening, the Expression of the CaG6PDH1 Is Downregulated Whereas CaG6PDH2 and CaG6PDH4 Are Upregulated. Exogenous NO Gas Exerts a Positive Modulation of CaG6PDH1, CaG6PDH3 and CaG6PDH4, but Negative of CaG6PDH2

Previous studies have shown that exogenous application of NO delays the pepper fruit ripening. This effect has been associated with other modifications at the biochemical and transcriptomic levels. Thus, NO triggered an increase in the content of ascorbate and upregulation of the activity and gene expression of the L-galactono-1,4-lactone dehydrogenase (GalLDH), the enzyme that catalyzes the last step of the ascorbate biosynthesis [44]. Furthermore, NO also modulated the metabolism of hydrogen sulfide (H_2_S) since the cytosolic L-cysteine desulfhydrase and mitochondrial D-cysteine desulfhydrase involved in its generation were positively regulated by NO [45]. Likewise, during pepper ripening, there is a prevalence in the superoxide radical (O_2_^•–^) generation by the NADPH oxidase (Rboh), which is negatively modulated in the presence of a NO-enriched environment [46].

However, to our knowledge, the regulation of the expression of the pepper *CaG6PDH* genes during fruit ripening and after treatment with NO is still unknown. This approach will provide a wider view of how the G6PDH system participates in the redox homeostasis that prevails during ripening under physiological conditions. The differential expression analysis of the four *CaG6PDH* genes indicated that, while the cytosolic *CaG6PDH1* was negatively modulated, the plastid *CaG6PDH2* and *CaG6PDH4* increased, and the plastid *CaG6PDH3* did not seem to be affected. Probably these effects could be associated with their subcellular locations considering that, during pepper fruit ripening, the chloroplasts undergo a dismantling of their internal membranes and are converted into chromoplasts, with concomitant photosynthesis downregulation and an accumulation of carotenoids.

Regarding G6PDH activity, our data indicate that this enzyme decreased at ripening, which is in agreement with previous reports, where the opposite behavior was also found in other NADP-dehydrogenases, including 6-phosphogluconate dehydrogenase (6PGDH), NADP-isocitrate dehydrogenase (NADP-ICDH), and NADP-malic enzyme (NADP-ME), which were increased [33,34]. This suggests that the contribution of NADPH that is not generated by G6PDH is compensated by the rest of NADP-dehydrogenases to maintain the redox homeostasis during ripening [34].

In other fruits, the relevance of G6PDH activity in response to different situations has been observed. Thus, G6PDH activity increased in banana fruit (*Musa* spp. Cavendish) under chilling conditions [47] or during postharvest storage of fruits such as apples or strawberries [48,49].

### 3.3. The Activity of the CaG6PDH Isozymes Is Inhibited by Tyr-Nitration, but Has a Complex Differential Regulation by Thiol-Based Oxidative Posttranslational Modifications (OxiPTMs)

To understand the possible mechanisms by which pepper fruit G6PDH could be modulated, the incubation of crude extracts from green fruits with a battery of regulatory compounds was assayed. It is known that the chloroplastic G6PDH isozymes are usually sensitive to redox regulation by NADPH and other redox compounds such as dithiothreitol or thioredoxin *m* [11,50,51]. Thus, during photosynthesis, those G6PDHs are usually inhibited since the source of NADPH production comes from the FNR at the photosystem I site. Nevertheless, cytosolic G6PDH is less sensitive to this regulation [52]. Furthermore, it has been reported that the activity of some G6PDH isozymes is inhibited under high NADPH concentrations, which is reasonable considering that the redox balance of the NADPH/NADP pool is highly modulated. In fact, it was proved that the plastidial G6PDH was inactivated via dithiol-disulfide protein interaction in green tissues from rice (*O. sativa*) [53].

NO and derived molecules, as well as H_2_S, can mediate PTMs which are able to affect the protein function. Among these PTMs, protein tyrosine nitration, *S*-nitrosation, and persulfidation should be mentioned. Protein Tyr-nitration is mediated by ONOO^−^, which usually causes a loss of function of the target protein [54]. There are numerous examples of plant enzymes that undergo Tyr-nitration, thus triggering their inhibition. They include the antioxidant enzymes catalase [55], ascorbate peroxidase, monodehydroascorbate reductase, and class III peroxidase from pepper, superoxide dismutase isozymes from Arabidopsis [56], and also some NADPH-generating dehydrogenases such as pea and pepper NADP-ICDH [57,58] and pepper and Arabidopsis NADP-ME [59,60]. To our knowledge, there are no data indicating that G6PDH could be inhibited by nitration. However, it is reasonable to think that, under a ONOO^−^ overproduction, the two G6PDH isozymes from pepper fruit could be significantly affected, as other NADP-generating enzymes are. Our data did prove that peroxynitrite (SIN-1 assays) inhibited both CaG6PDH I and CaG6PDH II. On the other hand, it was remarkable that, when the NO donors CysNO and GSNO, which simultaneously release Cys and GSH, respectively, were used, the activity of both G6PDH isoenzymes was less affected. This suggests that there is an *S*-nitrosation process that exerts a protective effect against the inhibition triggered by L-Cys on both isoenzymes and by GSH against CaG6PDH I. Similar behavior has been described in other enzymes from pepper fruits such as the H_2_S-generating cytosolic L-cysteine desulfhydrase and mitochondrial D-cysteine desulfhydrase [45], and the class III peroxidase. Therefore, it could be highlighted that pepper G6PDHs have a dual regulation by RNS; whereas ONOO^−^ triggers their inhibition by Tyr-nitration, the S-nitrosation exerts a positive role in protecting against the negative effects of GSH and L-Cys. This is a new aspect that should be considered particularly under adverse environmental conditions where the equilibrium of these molecules (NO, ONOO^−^, and GSH) is usually undertaken.

On the other hand, the data obtained indicate that both G6PDH isoenzymes were inhibited at 100% by H_2_S. This suggests that both would be susceptible to undergoing persulfidation, a PTM mediated by H_2_S [60]. Likewise, L-Cys exerted an inhibitory effect of 90% on both isoenzymes, while the tripeptide GSH (Glu-Cys-Gly) only did on CaG6PDH I, as indicated above. The thiol group of the cysteines that may putatively be involved in these effects can undergo different types of thiol-based oxidative posttranslational modifications (OxiPTMs), including *S*-nitrosation, persulfidation, *S*-glutathionylation, *S*-sulfenylation, *S*-cyanylation, and *S*-acylation, all of them mediated by NO, H_2_S, GSH, H_2_O_2_, CN^−^ and fatty acid, respectively. All these PTMs compete with each other depending on the accessibility of the thiol group of the Cys residue capable of being oxidized [60]. Therefore, the final effect will depend on the relative abundance among these signaling molecules and their accessibility to the potential target thiol group of the Cys residues of the G6PDH present in a specific subcellular location of either plastids, cytosol, or peroxisomes.

## 4. Materials and Methods

### 4.1. Identification of the G6PDH Genes in Pepper (Capsicum annuum L.) and Chromosomal Location

Complementary methodologies were used to identify the different *G6PDH* genes in pepper. First, the pepper proteome was downloaded from the NCBI database (Assembly UCD10Xv1.1; BioProject PRJNA814299). The G6PDH amino acid sequences from *Arabidopsis thaliana* were downloaded from the UnirProtKB database and their sequences were used as a query to search for G6PDHs in the complete pepper proteome using the BLASTP tool. Second, the InterProScan 5 software [61] was used to search for proteins that conserved in their sequence the G6PDH NAD(P) binding domain (PF00479) and G6PDH C-terminal domain (PF02781).

Location coordinates of the identified *CaG6PDHs* in the pepper genome were obtained from the NCBI database.

### 4.2. Phylogenetic and Conserved Motif Analyses of G6PDH Protein Sequences

The identified G6PDH protein sequences from sweet pepper, and G6PDHs described in several plant species (Appendix A), were used to construct a phylogenetic tree. The alignment of G6PDHs was performed using the CLUSTALW method [62]. Then, the aligned sequences were subjected to MEGA11 [63] to perform an unrooted maximum likelihood phylogenetic tree with default parameters. The obtained phylogenetic tree was improved using the online tool iTOL [64]. Conserved motifs of CaG6PDHs were analyzed using the MEME tool [65] and visualized using TBtools software v1.108 [66]. The protein localization based on their amino acid sequences was predicted using WoLF PSORT [67].

### 4.3. Cis-Regulatory Elements Analysis of the CaG6PDH Genes

To predict putative promoter sequences of the identified *CaG6PDH* genes obtained from the NCBI Nucleotide database (https://www.ncbi.nlm.nih.gov/nucleotide/; accessed on 10 February 2023), 1500 bp upstream from the transcription start point of each gene were considered. These sequences were searched for possible cis-acting regulatory elements using the PantCARE tool [68]. These results were manually processed and visualized using the ‘Basic Biosequence View’ function of TBtools v1.108 software [66].

### 4.4. Pepper Fruits and Exogenous Nitric Oxide (NO) Gas Treatment

The type of pepper fruits used and the procedure followed to expose them to an atmosphere enriched in NO have been previously described [31,53]. Briefly, sweet pepper (*Capsicum annuum* L. cultivar Melchor, California-type) fruits were collected from plants grown in plastic-covered greenhouses (El Ejido, Almería, Spain). Fruits without any damage were chosen at three developmental stages: green immature (G), breaking point (BP1), and red ripe (R). The selected fruits were transported to the laboratory at room temperature and kept at a low temperature (about 7 °C ± 1 °C) for 24 h. To assess the impact of the exogenous NO gas treatment, two additional groups with 6 fruits each were prepared: fruits exposed to 5 ppm NO for 1 h (BP2 + NO) and another group used as a control that was not treated with NO (BP2 − NO). After 3 days at room temperature, all groups of fruits were cut into small cubes (5 mm/edge), frozen under liquid nitrogen, and stored at −80 °C until their analysis. This experiment was performed thrice and the data shown are representative of the three replicates. Appendix A displays the experimental procedure used in this study with representative pictures of the phenotypes of pepper fruits at different ripening stages and subjected to NO treatment [31].

### 4.5. Library Preparation and RNA Sequencing

All procedures were performed as previously described in [31] with minor modifications. Briefly, libraries were prepared using an Illumina protocol and were sequenced on an Illumina NextSeq550 platform using 2 × 75 bp paired-end reads. These reads were preprocessed to remove low-quality sequences. Useful reads were mapped against the set of transcripts available for *Capsicum annuum* species in the NCBI database (assembly UCD10Xv1.1; accessed on 10 February 2023) using Bowtie2 [69]. Transcript counts were obtained using Samtools [70].

Differential expression analyses were completed using DEgenes-Hunter [71]. This R pipeline examined the relative change in expression between the different samples using different algorithms (EdgeR, DESeq2, Limma, and NOISeq) which apply their own normalizations and statistical tests to validate the whole experiment. On the other hand, an analysis of the time course of the *CaG6PDHs* gene expression was performed considering as a reference the expression levels found in green fruits (G). Raw data are accessible at the Sequence Read Archive (SRA) repository under the accession PRJNA668052. References to the pepper fruit transcriptome and differentially expressed (DE) genes among the analyzed ripening stages and the NO treatment involved the analysis of 24 biological replicates: 5 replicates for each stage, except for green fruits which involved 4 replicates.

### 4.6. Protein Modeling of CaG6PDHs

Using the artificial intelligence (AI) program AlphaFold [72], the monomeric tree-dimensional (3D) structure of CaG6PDHs was predicted. The residues involved in the Rossmann fold, G6P-binding site, and NADP-binding site are highlighted, as well as the two cysteine residues conserved among plastid CaG6PDH isoenzymes, and visualized using RasTop software (RasTop.Copyrights) (http://web.uni-plovdiv.bg/stu0704081022/User_2/Desktop/rastop2.2-source/rastop/help/copyright.htm (accessed on 6 February 2023).

### 4.7. Fruit Extracts, Protein Assay, Protein Enrichment, and In-Gel Isozyme Profile of G6PDH Activity

The pericarps of the fresh green pepper fruits were manually cut into small pieces of approximately 0.5 cm^2^. Then, samples were homogenized with a mortar and pestle in the presence of the extraction buffer in proportions 1:1 (*w*/*v*). The extraction buffer contained 50 mM Tris-HCl, 0.1 mM EDTA, 1 mM MgCl₂, 0.1% (*v*/*v*) Triton X-100, and 10% (*v*/*v*) glycerol. After homogenization, the extracts were filtered through two layers of nylon and centrifuged at 30,000× *g* for 20 min at 4 °C. Protein concentration was determined using the Bio-Rad protein assay (Hercules, CA), with bovine serum albumin as standard.

Protein enrichment was carried out as previously described [56]. Briefly, a solid powder of ammonium sulfate [(NH_4_)_2_SO_4_] was slowly added to the previously obtained 30,000× *g* supernatant of crude green pepper extracts up to a saturation of 50%. The solution was kept for approximately 30 min at 4 °C, then centrifuged at 30,000× *g* for 30 min, and the supernatant was 75% saturated via the slow addition of solid (NH_4_)_2_SO_4_. The solution was centrifuged again, and the protein-enriched pellet obtained (50–75% fraction) was suspended in 0.1 M Tris-HCl buffer, pH 8.0, 1 mM EDTA, 0.1% (*v*/*v*) Triton X-100, 10% (*v*/*v*) glycerol in a 12:1 ratio (for every 12 mL of original pepper extract the corresponding pellet was suspended in 1 mL). The 50–75% protein fraction was loaded on a Sephadex™ G-25 column which allows the separation of high and low molecular weight substances through desalting and buffer exchange.

Non-denaturing PAGE was carried out in 6% acrylamide gels (19:1, acrylamide:bis-acrylamide ratio) using a Mini-Protean Tetra Cell (Bio-Rad, Hercules, CA, USA). Pepper fruit crude extracts were added to 0.006% (*w*/*v*) bromophenol blue dye and then loaded onto gels. The G6PDH isozymes were visualized by incubating the gels in a solution consisting of 50 mM Tris-HCl, pH 7.6; 0.8 mM NADP; 5 mM EDTA; 2 mM MgCl_2_; 0.24 mM nitroblue tetrazolium; and 65 mM phenazine methosulfate containing 10 mM glucose-6-phosphate (G6P). When blue formazan bands appeared, the reaction was stopped by immersing the gels in 7% (*v*/*v*) acetic acid [8].

## 5. Conclusions

Early studies in sweet pepper fruits demonstrated the presence of G6PDH activity as well as its modulation during ripening [55]. However, to our knowledge, no information on the coding of genes for *G6PDH* in pepper, or on how they are expressed in fruits and modulated during ripening, is available. Therefore, the present study targeted those aims. The data shown in this work demonstrate that pepper fruits express four *CaG6PDH* genes (*CaG6PDH1-CaG6PDH4*) and two G6PDH isozymes (G6PDH I and G6PDH II) that are differentially regulated during ripening. Furthermore, the activity of CaG6PDH I and II isozymes underwent modulation by various oxiPTMs, among them the 100% inhibition of both isozymes that was exerted by H_2_S. Likewise, Tyr-nitration also provoked an inhibitory effect. These data provide new insights into the involvement of G6PDH in the ripening of pepper fruits as well as the interrelationship between the metabolism of NADPH, NO, and H_2_S. This information is of special interest for future research on the regulation of the NADPH levels of pepper fruits, attributing a central role to G6PDH in the NADPH-driven redox homeostasis during ripening and post-harvest quality control.

## Figures and Tables

**Figure 1 plants-12-03408-f001:**
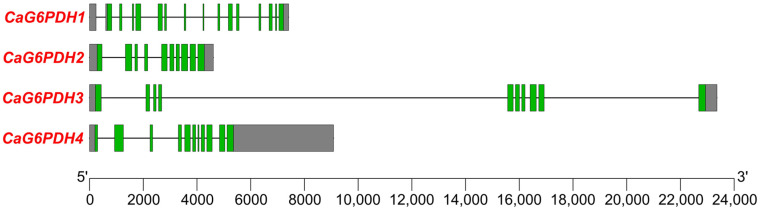
Genomic organization of the *CaG6PDH* gene family. The gene structure is displayed with exons (green boxes) and introns (black lines). Untranslated regions are shown in grey boxes. Exon–intron regions are drawn at scale.

**Figure 2 plants-12-03408-f002:**
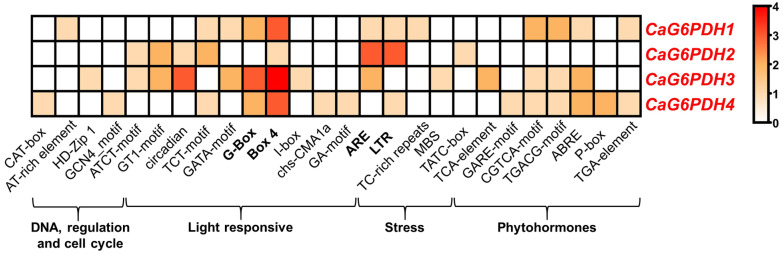
Heatmap of cis-regulatory elements corresponding to the 1500 bp upstream regions of *CaG6PDH* genes. The cis-regulatory elements were assembled according to their functional implications including DNA, regulation and cell cycle, light responsive, stress, and phytohormones. Motifs were identified in the PlantCARE database.

**Figure 3 plants-12-03408-f003:**
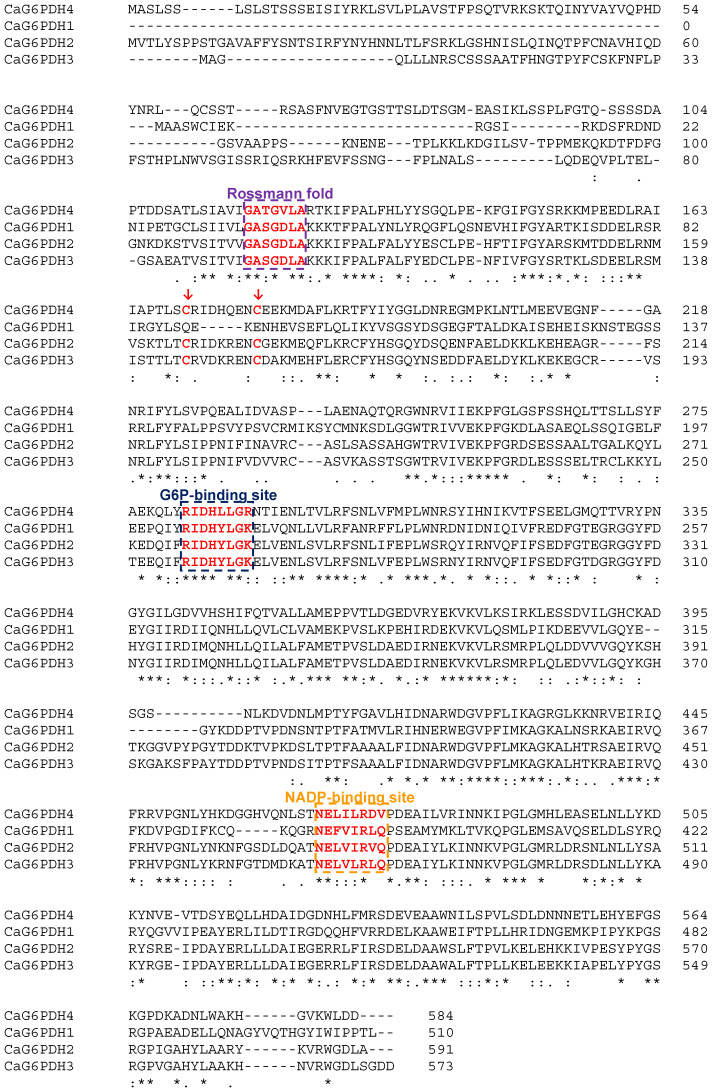
Multiple alignment of protein sequences of pepper (*Capsisum annuum* L.) CaG6PDHs. Conserved areas of the G6P and NADP binding sites, and the Rossmann fold are indicated by dashed squares. The two cysteine residues conserved among plastid CaG6PDH isozymes are pointed to by red arrows. The asterisk (*) indicates positions that have fully conserved residue. The colon (:) denotes conservation between residues with strong similar properties. The dot (·) indicates conservation between residues with weakly similar properties.

**Figure 4 plants-12-03408-f004:**
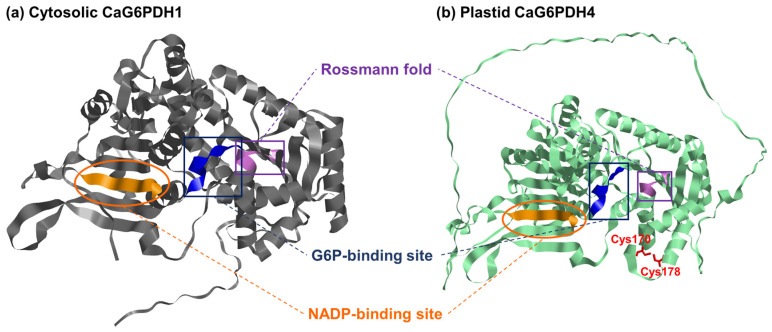
Model of the predicted tertiary structure of (**a**) the cytosolic CaG6PDH1 and (**b**) the plastid CaG6PDH4 from pepper fruits. The sheet NADP-biding domain, the Rossmann fold, as well as the G6P-binding site in the respective protein structures, are labeled with different colors. The two cysteine residues conserved among plastid CaG6PDH isozymes are depicted in red ink.

**Figure 5 plants-12-03408-f005:**
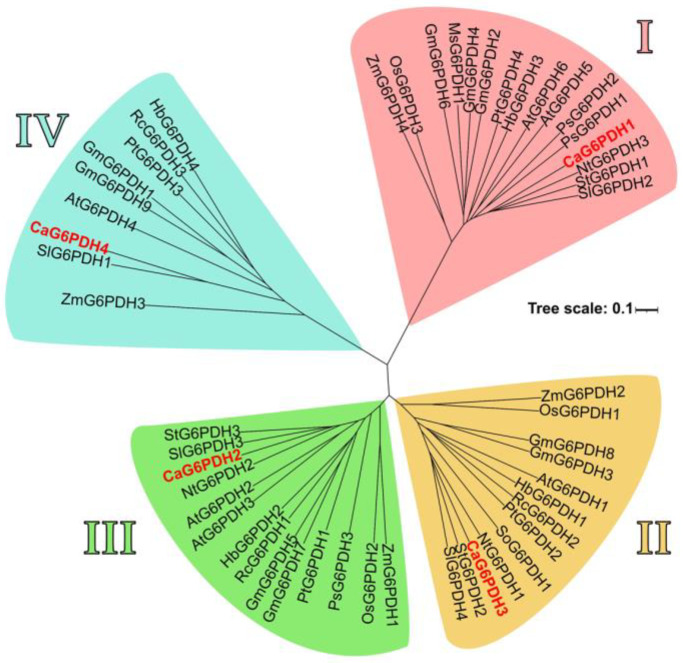
Phylogenetic tree of 53 G6PDHs from 15 different plant species. Clusters (**I**–**IV**) are displayed using different colors. Group (**I**) contains the cytosolic CaG6PDHs, while plastid CaG6PDHs are framed within groups (**II**–**IV**). The G6PDHs found specifically in the sweet pepper fruit are indicated in red. Species abbreviations: At, *Arabidopsis thaliana*; Ca, *Capsicum annuum*; Gm, *Glycine max*. Hb, *Hevea brasiliensis*; Ms, *Medicago sativa*; Nt, *Nicotiana tabacum*; Os *Oryza sativa*; Pc, *Petroselinum crispum*; Pt, *Populus trichocarpa*; Rc, *Ricinus communis*; Sl, *Solanum lycopersicum*; So, *Spinacia oleracea*; St, *Solanum tuberosum*; Ta, *Triticum aestivum*; and Zm, *Zea mays*.

**Figure 6 plants-12-03408-f006:**
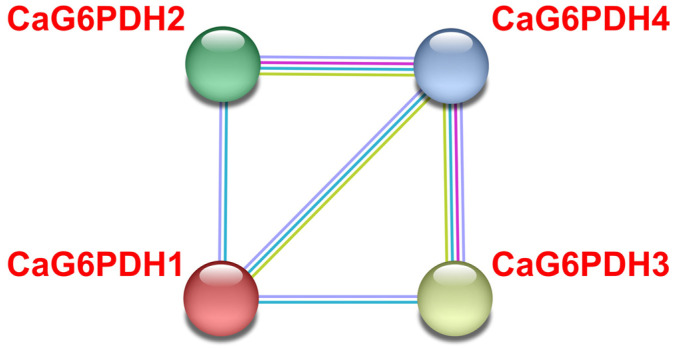
Predicted computational protein–protein interaction (PPI) network among the four identified CaG6PDHs. The color code of depicted lines is as follows: blue, known interactions from curated database evidence; green, neighboring genes; dark blue, gene co-occurrence; and purple, protein homology. The analysis was performed using STRING v11.0 with a minimum required interaction score set to “medium confidence” (0.400).

**Figure 7 plants-12-03408-f007:**
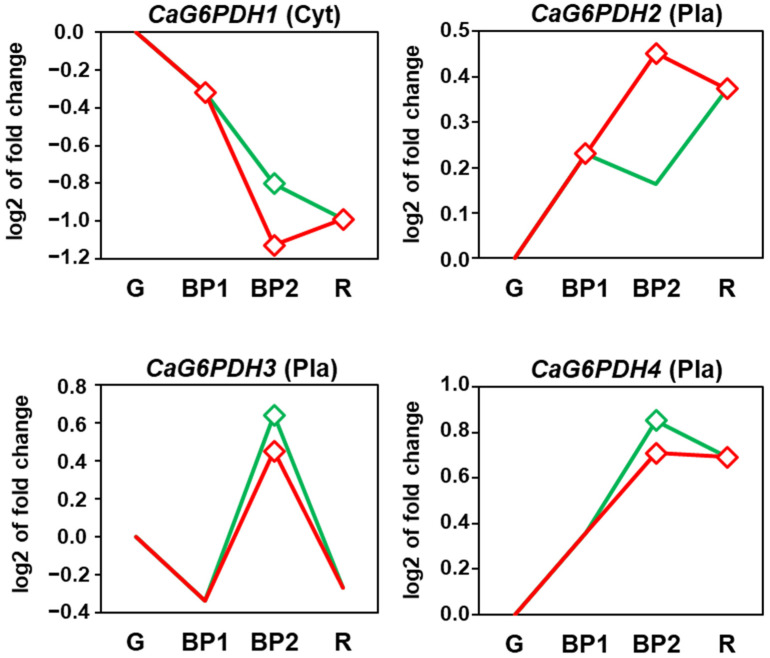
Time-course expression analysis (RNA-Seq) of four *CaG6PDH* genes under natural ripening conditions and after exogenous NO treatment. Samples of sweet pepper fruits at different ripening stages correspond to immature green (G), breaking point 1 (BP1), breaking point 2 with and without NO treatment (BP2 + NO and BP2 − NO, respectively), and ripe red (R). Diamonds indicate statistically significant changes in expression levels (*p* < 0.05) in comparison to immature green fruits (G). Green line: BP2 fruits treated with NO. Red line: untreated fruits.

**Figure 8 plants-12-03408-f008:**
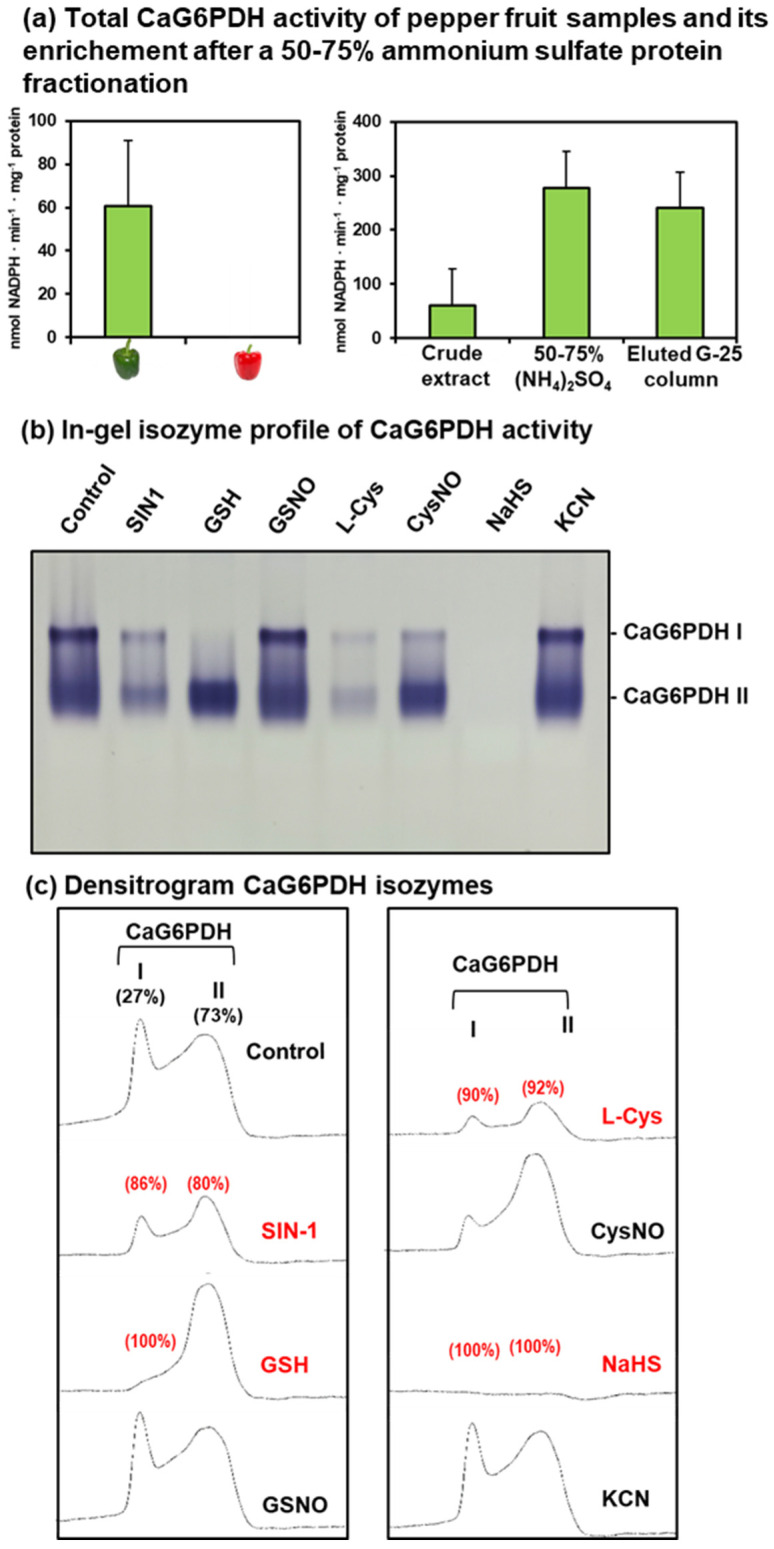
Total and isoenzymatic G6PDH activity of sweet pepper fruits after their enrichment through a 50–75% ammonium sulfate protein fractionation, and the effect of nitration, *S*-nitrosation, and reducing agents on the G6PDH isozymes. (**a**) Spectrophotometry analysis of total CaG6PDH activity in green and red pepper fruits and their enrichment after a 50–75% ammonium sulfate protein fractionation and further elution through a G-25 column. (**b**) In-gel isozyme profile of G6PDH activity in 6% acrylamide gels incubated with different modulating compounds. (**c**) Densitometric analysis of G6PDH isozymes after the treatment of green pepper fruit samples with different modulating compounds, and their relative quantification (%) made using the ImageJ program. SIN-1 is a peroxynitrite donor and a nitrating compound. GSNO (S-nitrosoglutathione) and CysNO (nitrosocysteine) are NO donors and nitrosating agents. L-Cys, cysteine. GSH, reduced glutathione. NaHS, sodium hydrosulfide as H_2_S donor. KCH, potassium cyanide. All treatments were carried out by pre-incubating the green pepper samples (26 µg protein per lane) with these compounds (5 mM) at 25 °C for 1 h, except for SIN-1, which was pre-incubated at 37 °C for 1 h. The number (in red) assigned to each peak indicates the percentage of inhibition provoked in the isozyme activities in comparison to the control samples (green fruit crude extracts).

**Table 1 plants-12-03408-t001:** Outline of the identified *G6PDH* genes in the pepper (*Capsicum annuum* L.) genome and the molecular properties of the protein encoded for these genes, including the chromosome (Chr.) location, the number of amino acids (aa), molecular mass (kDa), theoretical pI, and their subcellular localization.

Gene Name	Loc ID	Chr.	Genomic Location	Protein ID	Number of aa	Size (kDa)	Theoretical pI	Subcellular Localization
*CaG6PDH1*	107860735	2	153996151-154002712	XP_016561688.1	510	58.5	6.27	Cytosol
*CaG6PDH2*	107868504	4	771900-775889	XP_016570698.1	591	67.2	8.69	Plastid
*CaG6PDH3*	107878100	7	177247493-177247493	XP_016580465.1	573	65.2	6.36	Plastid
*CaG6PDH4*	107871175	8	156273784-156278939	XP_016573493.1	584	65.4	5.89	Plastid

## Data Availability

Sequence Read Archive (SRA) data are available at the following link https://www.ncbi.nlm.nih.gov/sra/PRJNA668052 (accessed on 28 May 2020).

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
