# Peer review of "In Silico RNAseq and Biochemical Analyses of Glucose-6-Phosphate Dehydrogenase (G6PDH) from Sweet Pepper Fruits: Involvement of Nitric Oxide (NO) in Ripening and Modulation"

_plants, 2023, doi:10.3390/plants12193408_

Round 1

Reviewer 1 Report

Reviewer’s Recommendation: Major Revision

Reviewer’s comments to Authors

Authors should revise the manuscript carefully in light of below comments...............

1.   Grammatical errors are present, please revise the whole manuscript to remove any possible grammatical and typos errors.

2.   Error in sentence formation, please revise the whole manuscript to avoid the use of long sentences and confusing sentences/paragraphs.

3.   Please maintain uniformity while in-text citation and referencing in the entire manuscript.

4.   The reference does not meet the format requirements of the Journal so please check the references as per the authors guideline of the Journal.

5.   It is advised to check and avoid too many self-cited papers. Authors are advised to cite maximum two self-papers.

6.   The beginning of a new paragraph should be after some space, check in complete manuscript.

7.      Throughout the whole manuscript the plant names should be in italic format.

8.      This paper lacks final revision by the author as many general repetitions, typos, grammatical, sentence formation errors were found in the manuscript. It is not possible to mention all such errors. Thus revise the manuscript accordingly.

Abstract:

Authors should revise the manuscript carefully in light of below comments...............

  1. An abstract must be fully self-contained and make sense by itself, without further reference to outside sources or to the actual paper. It is important to provide the relevance or importance of your work and the main outcomes. Please revise the abstract accordingly.
  2. The abstract is not clear and the objective of the paper is not clearly validated from the abstract.
  3. The future perspective of the experiment should be mentioned in the abstract.
  4. The abstract should appropriately over the contents of the manuscript.
  5. In the keywords, it is strongly advisable to use suitable words that can aid in finding out the manuscript in current registers or indexes. Strictly avoid the use of title words in the keywords.

6.      A graphical abstract is recommended for better perception of the present study.

7.      A novelty statement is also encouraged to be added in the manuscript for bringing out the uniqueness of your study and its importance.

8.      Please ensure that “Highlights” of the present work should be clearly mentioned in the manuscript.

Introduction:

Authors should revise the manuscript carefully in light of below comments...............

1.   The literature from past work done in the same field missing to strengthen the introduction section. The need and importance of the present work should be clearly written in the introduction section.

2.   The new aspects and innovations of this manuscript should be clearly and briefly described in this section.

3.   The present state of knowledge in the subject should be described in introduction.

4.   The literature should be sufficiently critical, current, and internationally evaluated.

Materials and Methods:

Authors should revise the manuscript carefully in light of below comments...............

1.         Please try to merge the different sub-sections of the methodology as an individual mention for each component seems a bit unscientific method.

2.         The size of manuscript seems to be large. It should be crisp and appropriate. Please revise it.

3.         The text presented across the manuscript should be simple so that the scientist/workers in other disciplines will understand. Please revise it.

4.         The different sections of manuscript are poorly cited with the references and required to update and validation with previous studies. The relevant papers listed below may be considered to enhance the scientific quality of manuscript significantly.

·         Gupta, P. et al. (2022). 24-Epibrassinolide Regulates Functional Components of Nitric Oxide Signalling and Antioxidant Defense Pathways to Alleviate Salinity Stress in Brassica juncea L. cv. Varuna. Journal of Plant Growth Regulation, pp.1-16. https://doi.org/10.1007/s00344-022-10884-y

·         Prajapati, P. et al. (2023). Nitric oxide mediated regulation of ascorbate-glutathione pathway alleviates mitotic aberrations and DNA damage in Allium cepa L. under salinity stress. International Journal of Phytoremediation25(4), pp.403-414. https://doi.org/10.1080/15226514.2022.2086215

·         Gupta, P. et al. (2020). Interactive role of exogenous 24 Epibrassinolide and endogenous NO in Brassica juncea L. under salinity stress: Evidence for NR-dependent NO biosynthesis. Nitric Oxide97, 33-47.https://doi.org/10.1016/j.niox.2020.01.014

Results and Discussion:

Authors should revise the manuscript carefully in light of below comments...............

  1. The results and discussion section needs to be elaborated more. The results should be clearly described in light of available knowledge and hypothesis and must be strongly validated with previous reports in the related subject area.
  2. The non-significant results was not clearly validated from the previous papers.
  3. Please carefully check, verify, and correct the results of the present experiments from the tables/figures/graphs provided in the manuscript.
  4. The discussion does not describe the results with proper facts and even does not validate the result with appropriate references. Please enrich it significantly.
  5. The discussion did not provide a specific reasons for the results. The provided explanation should be strengthen significantly.
  6. The strong hypothesis, scientific facts, and validation of previous reports are entirely missing. Please revise it. 

Conclusion:

Authors should revise the manuscript carefully in light of below comments...............

  1. The conclusion section failed to enlighten the spirit of the present finding or work so required to revise it accordingly.
  2. In the conclusion section the authors have only mentioned the data but major finding is missing from the conclusion part. Need to revise and incorporate this important concern of reviewer.
  3. The conclusion section seems like abstract so there is a need to revise the conclusion part accordingly.

Figures and Tables:

Authors should revise the manuscript carefully in light of below comments...............

·         Please provide the clear figures and tables.

·         The authors should write the descriptive, elaborated legends for the figures and the tables.

·         Please remove the redundancy from the legends of the figures and tables.

·         The legends of the figures and tables are not crisp and not completely bringing out the sense of the figures and tables. Rewrite it accordingly.

·         The placement of tables and figures in the manuscript should be done appropriately, which is missing in this manuscript. Please revise it.

·         The figures are overlapping the legends, the editing needs to be done.

·         The proper explanation of statistical analysis and its importance for describing the results should be mentioned.

·         There should not be monotony in representation of the results for instance all should not be represented in bar graph form vise-versa.

Reviewer 2 Report

You should pay attention to the spaces between words (Line 449,450,451).

In paragraph 4.4, it is not stated how many fruits you had per treatment and whether there were repetitions.

Author Response

Reviewer #2
You should pay attention to the spaces between words (Line 449,450,451).
Reply: Thanks for the comment. This refers to a website and the formatting processor does not allow other formula.
In paragraph 4.4, it is not stated how many fruits you had per treatment and whether there were repetitions.
Thank you. This has been addressed in the new version.

Round 2

Reviewer 1 Report

This reviewer appreciates the authors efforts to revise the manuscript. However, I do not convince with the present form of the manuscript. Therefore, this reviewer would encourage the authors to revise the manuscript significantly and may submit as a fresh submission in this journal. 
